# Initial Serum Levels of Magnesium and Calcium as Predictors of Mortality in Traumatic Brain Injury Patients: A Retrospective Study

**DOI:** 10.3390/diagnostics13061172

**Published:** 2023-03-19

**Authors:** Ahammed Mekkodathil, Ayman El-Menyar, Suhail Hakim, Hisham Al Jogol, Ashok Parchani, Ruben Peralta, Sandro Rizoli, Hassan Al-Thani

**Affiliations:** 1Clinical Research, Trauma and Vascular Surgery, Hamad Medical Corporation, Doha P.O. Box 3050, Qatar; 2Clinical Medicine, Weill Cornell Medical College, Doha P.O. Box 24144, Qatar; 3Trauma Surgery Section, Hamad General Hospital (HGH), Doha P.O. Box 3050, Qatar

**Keywords:** trauma, brain injury, head, electrolytes, calcium, magnesium, mortality

## Abstract

**Background:** We sought to evaluate the predictor role of the initial serum level of calcium and magnesium in hospitalized traumatic brain injury (TBI) patients. **Materials and methods:** A retrospective analysis of all TBI patients admitted to the Hamad Trauma Center (HTC), between June 2016 and May 2021 was conducted. Initial serum electrolyte levels of TBI patients were obtained. A comparative analysis of clinical variables between patients with abnormal and normal serum electrolyte level was performed. Logistic regression analysis with the variables that showed a significant difference (*p* < 0.05) in the bivariate analysis was performed to calculate the odds ratios (OR) for mortality. **Results:** There was a total of 922 patients with clinical records of serum electrolyte levels at admission. Of these, 757 (82.1%) had hypocalcemia, 158 (17.1%) had normal calcium level, and 7 (0.8%) had hypercalcemia. On the other hand, 616 (66.8%) patients had normal magnesium level, 285 (30.9%) had hypomagnesemia, and 12 (1.3%) had hypermagnesemia. The mortality rate in hypocalcemia group was 24% while in patients with normal calcium level it was 12%, *p* = 0.001. Proportionate mortality rates in hypomagnesemia and normal magnesium groups were 15% and 23% (*p* = 0.006), respectively. On the other hand, 7 out of 12 (58%) hypermagnesemia patients died during the index hospitalization. The regression model including GCS, ISS, PT, aPTT, INR, Hemoglobin, Bicarbonate, Lactate, Sodium, Potassium, Calcium, Magnesium, and Phosphate showed that hypocalcemia was not a significant predictor [OR 0.59 (CI 95%: 0.20–1.35)] of mortality after TBI. However, hypermagnesemia was a significant predictor [OR 16 (CI 95%: 2.1–111)] in addition to the GCS, ISS, aPTT, Bicarbonate, and Lactate values on admission. **Conclusion:** Although hypocalcemia and hypomagnesemia are common in hospitalized TBI patients, hypocalcemia was not a significant predictor of mortality, while hypermagnesemia was an independent predictor. Further studies with larger sample size and with prospective design are required to support these findings and their importance.

## 1. Introduction

Traumatic brain injury (TBI) remains as one of the most serious global public health challenges due to associated high case fatality, long-term disability, and socioeconomic burden [1,2,3,4]. Electrolyte imbalances in terms of abnormalities in serum sodium, potassium, calcium, and/or phosphate are common findings in TBI patients [5,6]. The most frequent cause of death following traumatic injury continues to be uncontrolled exsanguinating hemorrhage [7]. However, coagulopathy, hypothermia, and acidosis are critical trauma resuscitation factors constituting the lethal triad [8,9]. Recently, hypocalcemia was found to be interlinked with these factors and plays a key role in the outcomes of critically ill trauma patients, and therefore became the fourth component of a proposed lethal diamond [8,10]. 

The tight regulation of the coagulation cascade, which is essential for maintaining hemostasis, is mostly mediated by calcium ions. In addition to platelet activation, calcium ions also fully activate several other coagulation factors, such as coagulation Factor XIII. Clotting factor IV is a calcium ion that plays an important role in the intrinsic, extrinsic, and common pathways. Calcium is a divalent cation that can exist in several states, including a free, unbound, physiologically active state as well as an inert state that is attached to different proteins. While 55% of total calcium is bound to proteins (i.e., albumin) and citrate, only about 45% of it is physiologically active and resides in the ionized state. Derangements in the total body storage and serum can result from variations in the quantities of these proteins in the serum. Recent research has focused on hypocalcemia in trauma patients to improve resuscitation and comprehend the relationship between calcium derangements, mortality risk, and transfusion requirements. 

Magnesium is the fourth most abundant cation in the body and is largely present in bone (53%) and soft tissues (46%), with the remaining 1% being found in the blood, either in the free ionized state (54–65%) or attached to proteins (27–34%) and anions (8–12%) [11]. Due to its impact on several biomechanical processes, including neurotransmitter release, ion changes, oxidative stress, protein synthesis, and energy metabolism, magnesium plays a significant role in secondary brain injury [12]. The present study evaluated the predictor role of the initial serum level of calcium and magnesium in hospitalized TBI patients. We hypothesized that abnormal electrolytes levels on admission are associated with unfavorable outcomes after TBI. 

### Methods

A retrospective study was conducted in consecutive TBI patients admitted to the trauma intensive care unit (TICU) of the Hamad Trauma Center (HTC), Qatar, between 1 June 2016 and 30 May 2021. The HTC is the only level-1 trauma center in Qatar which treats moderate and severe injuries and at no cost. The study was based on the data obtained from the Qatar Trauma Registry (QTR) and the electronic medical records (CERNER). The QTR is a national level data repository consists of prospectively collected data on injury events, demographics, prehospital care, diagnoses, injury severity scores, provision of in-hospital care, and in-hospital outcomes of patients. The QTR has regular internal and external validation and reports to the national trauma databank (NTDB) and the American College of Surgeons Committee on Trauma’s (ACS- COT).

In this study, TBI was defined based on the ICD-10-CM (International Classification of Diseases, Tenth Revision, Clinical Modification) codes used in the QTR database. The codes included are S02.0, S02.1, S02.8, S02.9, S04.02, S04.03, S04.04, S06, S07.1, S09.90, which represent the cases with fracture of the skull, fracture of other specified skull and facial bones, unspecified fracture, injury of the optic chiasm, injury of optic tract and pathways, injuries of visual cortex, intracranial injury, crushing injury of skull, and unspecified injury of the head.

All patients, regardless of age or gender, diagnosed with TBI in the study duration, and provided their serum calcium or magnesium levels on admission to the ICU were available, were included in the study. The patients with penetrating injuries, and those transferred from other hospitals, were excluded. The patients were admitted to the trauma center and treated according to the guidelines of Advanced Trauma Life Support (ATLS). After the patients were stabilized, blood samples were drawn for arterial blood gases, blood chemistry, serum electrolytes (sodium, potassium, calcium, magnesium, and phosphate), and hematic biometry. Normal serum calcium and magnesium levels were defined as 2.2 to 2.7 mmol/L and 0.65 to 1.05 mmol/L respectively. Serum levels of calcium or magnesium below the lower limit of normal level were hypocalcemia or hypomagnesemia, and above the upper limit of normal level were hypercalcemia or hypermagnesemia.

The study collected demographic information of patients, mechanism of injuries, type of head trauma, injury scores, serum calcium levels on admission, other serum biomarkers levels on admission, interventions provided, complications developed in the hospital, and the in-hospital outcomes. The study was approved by the Institutional Review Board of Hamad Medical Corporation, Qatar (MRC#01-21-501)

Following a head injury, consciousness was evaluated using the Glasgow Coma Scale (GCS), which has a range of 3 to 15, and a severity scale of 4 to 8 for severe, 9 to 12 for moderate, and 13 to 15 for minor head injuries [13]. The Abbreviated Injury Scale (AIS) rates the severity of injuries inflicted on various body parts on a scale of 1 to 6, with minor (AIS = 1), moderate (AIS = 2), serious (AIS = 3), severe (AIS = 4), critical (AIS = 5), and non-survivable injuries (AIS = 6) [14]. The Injury Severity Score (ISS), which offers an overall score for polytrauma, is calculated by squaring the AIS ratings of the three most seriously injured body regions and adding them together [15]. The ISS scales from 0 to 75, with 1-8 being minor, 9–15 being moderate, 16–24 being serious, 50–74 being critical, and 75 being non-survivable.

## 2. Statistical Analysis

Demographic information, injury characteristics, complications in the hospital, and patient outcomes were presented as counts and percentages, mean and standard deviation, and median and interquartile range (IQR) whenever appropriate. The pattern of electrolyte levels based on TBI severity in terms of GCS was presented. 

Comparative analyses of clinical variables were performed between normal levels of calcium or magnesium and their abnormal levels. Comparison of categorical variables such as gender, mechanism of injuries, types of TBI, injured body regions, interventions, and complications between two groups were analyzed using chi-square test. On the other hand, comparison of mean values such as age, serum electrolyte levels, other biomarkers, and injury scores were performed using student *t*-test. Non-parametric tests were used to compare the medians and IQR of variables such as GCS, length of stays (LOS) on ventilator, in ICU, and in the hospital. The *p*-value < 0.05 was considered as statistically significant result for all these tests. Logistic regression analysis with the relevant clinical and laboratory variables that showed a significant difference (*p* < 0.05) in the bivariate analysis was also performed to calculate the odds ratios (OR) for mortality. The OR was reported with 95% confidence upper and lower limits. The data analyses were performed using the Statistical Package for the Social Sciences v21.0 (SPSS, Inc., Chicago, IL, USA).

## 3. Results

During the study period, there were 922 hospitalized TBI patients with clinical records of serum electrolyte levels at admission to the HTC. Of the total patients, 757 (82.1%) had hypocalcemia at admission, 158 (17.1%) had normal calcium level, and 7 (0.8%) had hypercalcemia. Serum magnesium levels were reported in 913 patients: 616 (67.5%) were normal, 285 (31.2%) were hypomagnesemia, and 12 (1.3%) were hypermagnesemia.

Our patient population was young (mean age 32 years) and predominantly male (94%). Most patients sustained road traffic injuries (59%), followed by falls (25%). The most common types of TBI were subarachnoid hemorrhage (SAH) and subdural hematoma (SDH); 42% and 35% respectively. The most common associated injuries were the thoracic (55%) and abdominal (27%) injuries. Head injuries were mostly severe as the mean head AIS was nearly 4, and critical or maximal injury (AIS ≥ 5) was 37%, while serious (AIS = 4) or severe AIS (AIS = 3) represented 58 percent. Forty-five (4.9%) patients had moderate head injuries with AIS = 2 and 4 (0.4%) patients had minor head injuries with AIS = 1. These patients with minor head injuries were treated in the ICU mainly because of the associated injuries. The median GCS of the patient population was 3, with severe (3–8), moderate (9–12) GCS in 681 (74%), and 166 (18%) patients, respectively. Mild GCS was found in 66 (7.2%) patients (Table 1). Figure 1 demonstrates the study design and proportion of electrolytes imbalance in TBI patients. Of note, 59 patients with normal calcium level had abnormal magnesium level, 487 patients with normal magnesium level had abnormal calcium level, and 269 patients had abnormal levels of both calcium and magnesium.

The pattern of serum calcium and magnesium levels by GCS was demonstrated in Table 2. Abnormal calcium levels were very common across all GCS categories, and this was highly prevalent in patients with severe GCS (*p* = 0.001). On the other hand, abnormal serum magnesium levels were less prevalent than normal levels in all GCS categories; however, hypomagnesemia was evident in almost one third of each GCS category, and hypermagnesemia was rare. Table 3 demonstrates the levels of serum electrolytes in different TBI lesions. SAH was more common in hypocalcemia and hypermagnesemia.

The overall in-hospital mortality rate was 22 percent. Data on proportionate mortality among normal, hyper, and hypo levels of serum calcium and magnesium showed the highest mortality rate in the hypercalcemia group (71.4%), followed by the hypermagnesemia group (58.3%). Proportionate mortality in hypocalcemia was 23.8%, almost twice when compared to the patients with normal calcium levels (12%) (*p* = 0.001). Mortality in hypomagnesemia was comparably lower than in patients with normal magnesium level (15.4% vs. 23.4%, *p* = 0.001).

Comparative analyses between normal levels and hypo levels of serum calcium and magnesium were performed. Although the mortality associated with hyper levels were higher, its prevalence was very low. SAH and SDH were more frequent in the hypocalcemia group, all other types of injuries were similar across calcium and magnesium groups. Normal and hypomagnesemia groups had similar GCS and head AIS; however, ISS was significantly higher in the hypomagnesemia group, whereas low GCS and high ISS was found in the hypocalcemia group when compared to the normal calcium group (Table 4).

Sodium and magnesium levels were found to vary significantly with variations in calcium levels, while calcium and phosphate levels fluctuate significantly with magnesium level variations (Table 5). The bivariate analysis further showed that the calcium level was also linked with other clinical variables, such as bicarbonate level, Hb, PT, aPTT, and INR. Magnesium levels were associated with lactic acid, PT, and INR. Poor outcomes of patients in terms of mechanical ventilator days, ICU duration of stay, and hospital LOS were significantly longer in the hypocalcemia group when compared to normal calcium levels and in the hypomagnesemia group when compared to normal magnesium levels. In-hospital mortality was significantly higher in hypocalcemia group (24% vs. 12%, *p* = 0.001) while it was lower in the hypomagnesemia group (15% vs. 23%, *p* = 0.006) when compared to their normal levels (Table 4).

**Predictors of mortality**: The logistic regression analysis included GCS, ISS, PT, aPTT, INR, Hg, Bicarbonate, Lactate, Sodium, Potassium, Calcium, Magnesium, and Phosphate levels (the clinical severity and laboratory variables on admission). The regression model showed five significant predictors of mortality as follows: (1) hypermagnesemia [OR 16.32 (CI 95%: 2.38–111.77)]; (2) higher ISS [OR 1.05 (CI 95%: 1.03–1.08)]; (3) prolonged thromboplastin time [OR 1.05 (CI 95%: 1.02–1.08)]; (4) lower GCS on admission [OR 0.89 (CI 95%: 0.82–0.96)]; and (5) lower serum bicarbonate levels [OR 0.93 (CI 95%: 0.87–0.99)]. Hypocalcemia was not a significant predictor [OR 0.52 (CI 95%: 0.20–1.36)] (Table 6).

## 4. Discussion

The present study examined the role of the initial serum levels of calcium and magnesium in predicting the mortality of hospitalized TBI patients. The study found that hypocalcemia and hypomagnesemia are common in TBI patients whereas hypercalcemia or hypermagnesemia is rare. Although abnormal calcium levels were common regardless of injury severity of patients in terms of GCS, it was more evident in severe TBI. Magnesium imbalance was less common than calcium imbalance in our study cohort.

Hypocalcemia was associated with increased in-hospital mortality while hypomagnesemia was not. However, hypocalcemia and hypomagnesemia patients had significantly longer duration of stay in the hospital. The logistic regression model concluded that hypocalcemia was not a significant predictor of mortality after TBI; however, hypermagnesemia was a significant predictor in addition to the GCS, ISS, aPTT, serum bicarbonate, and lactate initial values. Our findings on the role of magnesium and calcium levels in predicting TBI mortality and pattern of lesions could add to the existing literature and provide valuable insights into the complex interplay between electrolyte imbalances and TBI pattern and outcomes.

Multiple studies suggest that hypocalcemia is a common finding in trauma patients, which is associated with mortality, blood transfusion requirements, and coagulopathy [16,17,18]. Similarly, the role of the magnesium ion in maintaining brain homeostasis following TBI is widely accepted [12,19]. Multiple secondary injury mechanisms, including neurotransmitter release, ion alterations, oxidative stress, protein synthesis, and energy metabolism are impacted by magnesium concentration. Vasomotor tone, platelet function, and intrinsic and extrinsic pathway-mediated coagulation are all dependent on calcium-dependent pathways, and these pathways are essential for hemorrhagic shock and resuscitation. The bivariate analysis found that hypocalcemia was associated with lower bicarbonate level; this may be due to shared physiological pathways or mechanisms. However, hypomagnesemia was associated with lower lactate. Elevated levels of lactate in the bloodstream can lead to metabolic acidosis and hinder the brain’s capacity to control blood flow and sustain typical brain activity. This, in turn, can raise the likelihood of secondary brain injury and mortality.

Cherry et al. [16] conducted a retrospective study in 396 trauma patients who required highest level of response which investigated the relationship between ionized calcium (iCa) levels with injury severity, acidosis, hypotension, and mortality. The serum samples were drawn from the patients at the time of arrival to the emergency department, and the investigators categorized patients into two groups based on their iCa, i.e., iCa < or = 1 group versus iCa > 1 [16]. The study found that the mortality was significantly higher and time to death was significantly shorter in the iCa < or = 1 group. The study concluded that regardless of age, ISS, or sample period, low iCa is linked to prehospital hypotension and is a greater predictor of mortality than base deficit [16].

Magnotti et al. [17] collected iCa on-admission prospectively in 591 trauma patients and found that multiple transfusion, massive transfusion, and mortality was increased significantly in iCa < 1 group. After correcting for age and the severity of the injury, multivariable logistic regression analysis found iCa as an independent predictor of the need for multiple transfusions (OR = 2.29, 95% CI = 1.05–4.99) [17]. Admission iCa levels may make it easier to quickly identify patients who need massive transfusions, enabling early preparation and administration of proper blood products.

Vasudeva et al. [18] conducted a retrospective study which investigated the association between hypocalcemia (iCa measured on admission) and adverse outcomes in 113 shocked (shock index ≥ 1) major trauma patients. Patients with pre-hospital blood transfusion were not included. Acute traumatic coagulopathy was defined as an initial INR > 1.5, while ionized hypocalcemia was defined as <1.11 mmol/l. The study demonstrated that the admission iCa level was associated with blood transfusion requirement in the first 24-h post admission and death at hospital discharge. Notably, the previous studies discussed above were based on the ionized calcium value. Understanding whether free calcium or magnesium behave differently in TBI patients is critical and not well-defined.

Vinas-Rios et al. [20] conducted a comparative case control study to evaluate whether serum hypocalcemia on the third day [defined as <2.1 mmol/l (8.5 mg/dL)] is a prognostic factor for early mortality after moderate and severe TBI. The study included patients with aged between 1 and 89 years and had a GCS of 3–12 points following TBI. The study found that serum calcium level on the third day could be a useful predictor of mortality (OR = 3.5, 95% CI: 1.12 to 13.61, *p* < 0.028). In addition, the study reported that impaired pupillary reactivity was significantly associated mortality in TBI patients.

These findings were further established by Manuel et al. [21], who conducted a retrospective study of 99 patients with moderate and severe TBI diagnosed with intracranial lesions in cranial computed tomography (CT). In this study, serum hypocalcemia was defined as <2.1 mmol/L [8.5 mg/dL] and the hypocalcemia of iCa was defined as <1.10 mmol/L [4.5 mg/dL]. The study concluded that serum levels of iCa on the third day could be a useful predictor of mortality and disability. The ROC analysis demonstrated the cut-off value on the 3rd day was 1.11 mmol/L, with greater sensitivity (83.76%) and specificity (66.66%), with an OR 6.45 (95% CI: 2.02–20.55).

Increased risk of development of hypomagnesemia in TBI patients was evident from the current study. Polderman et al. [22] studied 18 severe head injury patients admitted to the ICU and compared with 19 trauma patients with no significant cranial trauma. In the severe head trauma patient group, 67% patients had hypomagnesemia while the control group had only 10% (*p* < 0.01). In this study, hypomagnesemia was defined as <0.70 mmol/L. In our study, hypomagnesemia was defined as <0.65 mmol/L. The study by Polderman et al. [22] reported hypomagnesemia, hypophosphatemia, and hypokalemia among severe head injury patients; however, the mortality was not studied. Mendez et al. [23] conducted a prospective study of 98 hospitalized pediatric TBI patients, which demonstrated that TBI in children was associated with a reduction in total magnesium level, whereas ionized magnesium level decreased only with severe TBI. According to the study, variations in plasma ionized magnesium levels could be used as a marker for TBI, but only for a short period of time.

Recently, Wang et al. [24] studied the relationship between early serum magnesium levels and mortality of TBI patients. Multivariate regression analysis showed that serum magnesium level was positively associated with mortality after adjusting the confounding factors. The study concluded that patients with TBI and abnormally low or high serum magnesium levels had an increased risk of mortality. In agreement with our findings, Wang et al. [24] concluded that higher initial serum magnesium level is independently associated with mortality in TBI patients.

One-third of the 14,101 ICU patients (886 trauma patients) studied by Laupland et al. [25] had abnormal magnesium levels at the time of admission. The most prevalent condition, hypomagnesemia, did not increase the risk of death. A twofold higher risk of death was linked to hypermagnesemia at the time of admission, although this association was only shown in patients who developed hypermagnesemia after being admitted to the ICU. Both hypomagnesemia and hypermagnesemia upon admission were not associated with death after adjusting for confounding factors. However, there was a higher risk of death when ICU-acquired hypermagnesemia developed in patients who had normal or low magnesium levels before admission.

## 5. Limitations

This retrospective study, in which we used ICD codes from the national trauma registry database to include TBI patients with different lesions, may entitle a selection bias. Moreover, the low sample size in each TBI lesion limited the proper comparative analyses, and further larger studies are required. The patient population did not comprise exclusively severely injured TBI cases, with about 25% having mild to moderate injury based on the GCS scores and less than 5% having head AIS scores below two. Polytrauma was common, with a mean ISS of 27 and chest injury being the most prevalent associated injury in over half of the patients. However, this group still represents a diverse range of TBI patients in our country. Our focus was on the initial serum electrolyte levels; therefore, their changes during ICU stay did not allow us to examine the time-dependent nature of calcium and magnesium on TBI outcomes. Previous research suggested that electrolyte levels can fluctuate during ICU stay. Hypocalcemia or hypomagnesemia, for example, often occurs from day 3 to day 5 and may persist for 2–3 weeks [26]. Finally, functional outcomes and cognitive status were not evaluated in this study as they fall outside the study scope.

## 6. Conclusions

Although hypocalcemia and hypomagnesemia are common in hospitalized TBI patients, hypocalcemia was not a significant predictor of mortality, while hypermagnesemia was an independent predictor. However, the latter was encountered in a small sample size. Further studies with larger sample size and with prospective design are required to support these findings and their importance after brain injury.

## Figures and Tables

**Figure 1 diagnostics-13-01172-f001:**
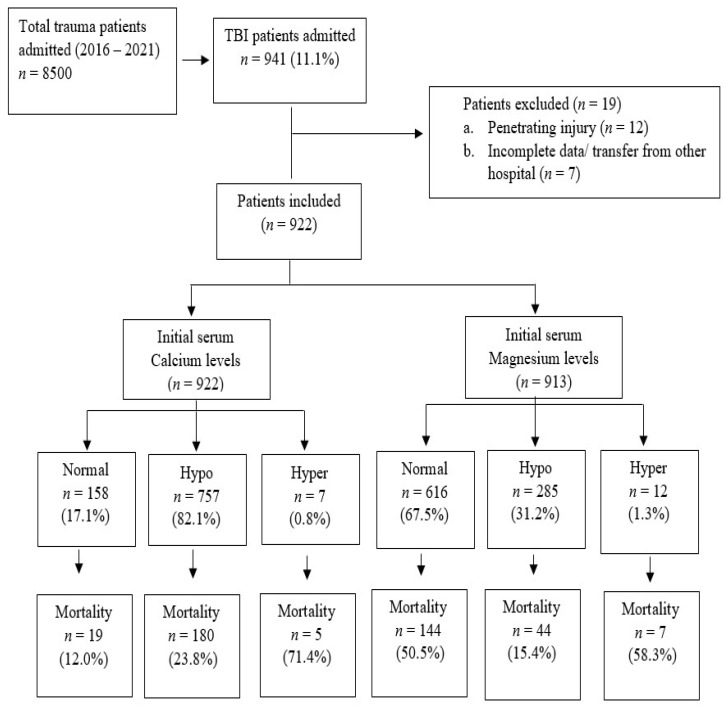
Study flow chart for electrolytes imbalance in TBI patients.

**Table 1 diagnostics-13-01172-t001:** Demographic and clinical variables of hospitalized traumatic brain injury patients (TBI) (*n* = 922).

Variable	Value
Age [mean± standard deviation (SD)]	32.2 ± 15.0
Males	863 (93.6%)
TBI types	
*Epidural hematoma*	204 (22.1%)
*Subdural hematoma*	321 (34.8%)
*Subarachnoid hemorrhage*	387 (42.0%)
*Compression of basal cisterns*	110 (11.9%)
*Effacement of Sulci*	171 (18.5%)
*Midline Shifts*	206 (22.3%)
Injury Severity Score [median (interquartile range, IQR)]	27 (18–34)
Glasgow Coma Scale [median (IQR)]	3 (3–9)
Shock index [median (IQR)]	0.8 (0.7–1.0)
Abbreviated Injury Scale (AIS) (mean ± SD)	
*Head AIS*	3.9 ± 0.97
*Chest AIS*	2.78 ± 0.70
*Abdomen AIS*	2.65 ± 1.0
*Cervical spine AIS*	2.26 ± 0.67
*Thoracic spine AIS*	2.16 ± 0.65
*Lumbar spine AIS*	2.01 ± 0.09
Laboratory findings [median (IQR)]	
*Initial serum Sodium*	141.0 (139–143)
*Initial serum Potassium*	3.8 (3.4–4.1)
*Initial serum Calcium*	2.0 (1.8–2.1)
*Initial serum Magnesium*	0.7 (0.6–0.8)
*Initial serum Phosphate*	0.9 (0.7–1.2)
*Initial serum Bicarbonate*	19.6 (16.7–23.0)
*Initial serum Lactic acid*	2.9 (2.0–4.3)
Prothrombin time [median (IQR)]	12.0 (11.1–13.5)
Activated partial thromboplastin time [median (IQR)]	26.2 (24.0–31.0)
International normalized ratio [median (IQR)]	1.1 (1.1–1.3)
Initial serum hemoglobin [median (IQR)]	13.0 (11.3–14.4)
Initial serum glucose [median (IQR)]	8.0 (6.7–10.1)
Intubation	827 (89.7%)
Massive transfusion protocol activation	138 (15.0%)
Ventilator associated pneumonia	121 (13.1%)
Mechanical ventilator days [median (IQR)]	5 (2–11)
Intensive care unit days [median (IQR)]	9 (4–17)
Hospital length of stay [median (IQR)]	17 (7–32)
In-hospital mortality	204 (22.1%)

**Table 2 diagnostics-13-01172-t002:** Calcium and magnesium imbalance in each GCS category in TBI patients.

	Mild GCS (13–15) (*n* = 88)	Moderate (9–12) (*n* = 144)	Severe (3–8) (*n* = 681)	Total
**Initial serum calcium (Ca) levels at admission**				
Normal	29 (33.0%)	49 (34.0%)	79 (11.6%)	157 (17.2%)
Hypocalcemia	59 (67.0%)	94 (65.3%)	596 (87.5%)	749 (82.0%)
Hypercalcemia	0	1 (0.7%)	6 (0.9%)	7 (0.8%)
**Total**	88 (100%)	144 (100%)	681 (100%)	913 (100%) *
**Initial serum magnesium (Mg) levels at admission**				
Normal	64 (72.7%)	97 (67.4%)	449 (66.8%)	610 (67.5%)
Hypomagnesemia	24 (27.3%)	46 (31.9%)	212 (31.5%)	282 (31.2%)
Hypermagnesemia	0	1 (0.7%)	11 (1.6%)	12 (1.3%)
**Total**	88 (100%)	144 (100%)	672 (100%)	904 (100%) *

* Few GCS data were missing.

**Table 3 diagnostics-13-01172-t003:** Electrolyte imbalance based on the type of Traumatic Brain Injury lesion.

	**Initial Calcium Imbalance**							
	**Normal (*n* = 158)**		**Hypercalcemia (*n* = 7)**		**Hypocalcemia (*n* = 757)**		**Total (*n* = 922)**	
**TBI Types**								
Epidural hemorrhage	41	25.9%	1	14.3%	162	21.4%	204	22.1%
Subdural hemorrhage	43	27.2%	1	14.3%	277	36.6%	321	34.8%
Subarachnoid hemorrhage	51	32.3%	2	28.6%	334	44.1%	387	42.0%
Compression of basal cisterns	18	11.4%	2	28.6%	90	11.9%	110	11.9%
Effacement of sulci	35	22.2%	4	57.1%	132	17.4%	171	18.5%
Midline shift	34	21.5%	1	14.3%	171	22.6%	206	22.3%
	**Initial Magnesium imbalance**							
	**Normal (*n* = 616)**		**Hypermagnesemia (*n* = 12)**		**Hypomagnesemia (*n* = 285)**		**Total (*n* = 913)**	
**TBI Types**								
Epidural hemorrhage	145	23.5%	1	8.3%	58	20.4%	204	22.3%
Subdural hemorrhage	205	33.3%	3	25.0%	111	38.9%	319	34.9%
Subarachnoid hemorrhage	250	40.6%	7	58.3%	128	44.9%	385	42.2%
Compression of basal cisterns	80	13.0%	3	25.0%	24	8.4%	107	11.7%
Effacement of sulci	112	18.2%	4	33.3%	54	18.9%	170	18.6%
Midline shift	136	22.1%	3	25.0%	67	23.5%	206	22.6%

**Table 4 diagnostics-13-01172-t004:** Comparative analysis between serum hypo levels and normal levels for calcium (N = 915) * and magnesium (N = 901) ** at admission in patients with traumatic brain injuries.

Variables	Normal Ca Level (*n* = 158, 17.3%)	Hypocalcemia (*n* = 757, 82.7%)	*p*-Value	Normal Mg Level (*n* = 616, 68.4%)	Hypomagnesemia (*n* = 285, 31.6%)	*p*-Value
Age	32.5 ± 17.2	32.2 ± 14.5	0.843	32.0 ± 15.3	32.5 ± 14.1	0.645
Males	144 (91.1)	713 (94.2)	0.153	580 (94.2)	264 (92.6)	0.382
Epidural hematoma	41 (25.9)	162 (21.4)	0.211	145 (23.5)	58 (20.4)	0.287
Subdural hematoma	43 (27.2)	277 (36.6)	0.025	205 (33.3)	111 (38.9)	0.097
Subarachnoid hemorrhage	51 (32.3)	334 (44.1)	0.006	250 (40.6)	128 (44.9)	0.221
Compression of basal cisterns	18 (11.4)	90 (11.9)	0.860	80 (13.0)	24 (8.4)	0.046
Effacement of sulci	35 (22.2)	132 (17.4)	0.163	112 (18.2)	54 (18.9)	0.076
Midline shifts	34 (21.5)	171 (22.6)	0.769	136 (22.1)	57 (23.5)	0.633
Intubation	121 (76.6)	699 (92.3)	0.001	537 (87.2)	269 (94.4)	0.001
Massive transfusion protocol	9 (6.7)	127 (16.8)	0.001	83 (13.5)	45 (15.8)	0.355
Craniotomy/craniectomy	30 (19.0)	160 (21.1)	0.545	120 (19.5)	70 (24.6)	0.082
Ventilator associated pneumonia	11 (7.0)	109 (14.4)	0.012	77 (12.5)	44 (15.4)	0.229
Glasgow Coma Scale	8 (3–12)	3 (3–8)	0.001	3 (3–9)	3 (3–8)	0.208
Head AIS	3.97 ± 0.97	3.91 ± 0.97	0.702	3.9 ± 0.9	3.9 ± 1.0	0.951
Injury Severity Score	22.8 ± 10.5	27.7 ± 10.2	0.001	26.3 ± 10.4	28.0 ± 10.5	0.024
Mechanical ventilator days	2 (1–7.5)	6 (2–12)	0.001	4 (1–11)	7 (3–12)	0.001
Intensive care unit days	6 (2–12)	8 (3–17)	0.012	7 (3–14)	10.5 (5–18)	0.001
Hospital length of stay in days	12 (5–22)	17 (7–31)	0.003	14 (6–28)	20.5 (10–35)	0.001
In-hospital mortality	19 (12.0)	180 (23.8)	0.001	144 (23.4)	44 (15.4)	0.006

* 7 patients with hypercalcemia excluded; ** 12 patients with hypermagnesemia excluded.

**Table 5 diagnostics-13-01172-t005:** Laboratory findings at admission in patients with traumatic brain injuries.

Variables	Normal Ca Level (*n* = 158, 17.3%)	Hypocalcemia (*n* = 757, 82.7%)	*p*-Value	Normal Mg Level (*n* = 616, 68.4%)	Hypomagnesemia (*n* = 285, 31.6%)	*p*-Value
Initial serum sodium	140.0 ± 4.0	141.3 ± 4.9	0.002	140.9 ± 4.5	141.2 ± 4.4	0.382
Initial serum potassium	3.8 ± 0.6	3.8 ± 0.7	0.175	3.82 ± 0.6	3.77 ± 0.6	0.273
Initial serum calcium	2.34 ± 0.23	1.90 ± 0.20	0.001	2.01 ± 0.26	1.90 ± 0.24	0.001
Initial serum magnesium	0.75 ± 0.13	0.69 ± 0.11	0.001	0.74 ± 0.07	0.59 ± 0.05	0.001
Initial serum phosphate	1.0 ± 0.4	1.0 ± 0.5	0.763	1.04 ± 0.46	0.93 ± 0.35	0.001
Initial serum bicarbonate	21.5 ± 4.5	19.0 ± 4.1	0.001	4.17 ± 0.16	3.95 ± 0.23	0.092
Initial serum lactic acid	3.8 ± 2.9	3.7 ± 2.7	0.680	3.80 ± 2.81	3.24 ± 1.88	0.001
Initial serum hemoglobin	13.7 ± 2.4	12.5 ± 2.3	0.001	12.80 ± 2.32	12.53 ± 2.42	0.120
Initial serum glucose	11.5 ± 4.9	8.5 ± 3.8	0.001	9.08 ± 3.96	8.71 ± 3.43	0.175
Prothrombin time	12.1 ± 3.3	13.8 ± 9.3	0.001	13.29 ± 6.76	12.60 ± 3.13	0.036
Activated partial thromboplastin time	29.3 ± 16.8	32.8 ± 23.0	0.030	31.39 ± 18.55	30.23 ± 16.77	0.369
International normalized ratio	3.8 ± 2.9	1.3 ± 0.8	0.001	1.27 ± 0.58	1.23 ± 0.28	0.039

**Table 6 diagnostics-13-01172-t006:** Multivariable logistic regression analysis for the predictors of mortality in Traumatic Brain Injury using the clinical severity and laboratory variables on admission.

	Odds Ratio	95% CI Lower	95% CI Upper	*p*-Value
Initial serum sodium	1.024	0.974	1.077	0.345
Initial serum potassium	1.112	0.792	1.563	0.539
Initial serum calcium	0.523	0.202	1.355	0.182
Initial serum bicarbonate	0.926	0.869	0.987	0.018
Initial serum phosphate	0.963	0.596	1.555	0.877
Initial serum magnesium	16.315	2.381	111.771	0.004
Initial serum hemoglobin	1.034	0.933	1.146	0.526
Injury severity score	1.052	1.028	1.077	0.001
GCS on admission	0.887	0.824	0.955	0.001
Prothrombin time	1.034	0.859	1.244	0.726
Activated partial thromboplastin time	1.049	1.022	1.077	0.001
International normalized ratio	2.34	0.294	18.632	0.422
Initial serum lactate level	1.156	1.053	1.269	0.002

## Data Availability

All data generated or analyzed during this study are included in this article. Data are accessible upon agreement with the Medical Research Centre, Qatar.

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
