# Peer review of "Initial Serum Levels of Magnesium and Calcium as Predictors of Mortality in Traumatic Brain Injury Patients: A Retrospective Study"

_diagnostics, 2023, doi:10.3390/diagnostics13061172_

Round 1

Reviewer 1 Report

This study investigated the role of calcium and magnesium levels in predicting the mortality of hospitalized traumatic brain injury (TBI) patients. Based on the results it is evident, that hypocalcemia and hypomagnesemia is common in TBI patients. Hypocalcemia was associated with an increased risk of in-hospital mortality, while hypomagnesemia was not, but both were linked to longer hospital stays. The logistic regression model suggested that hypermagnesemia was a significant predictor of mortality after TBI, along with other initial values. Multiple studies suggest that hypocalcemia is associated with mortality, blood transfusion requirements, and coagulopathy.

In general, the research yields significant and applicable findings that could potentially contribute to the forecasting of Traumatic Brain Injury (TBI) outcomes, subject to additional research. The amount and caliber of the referenced literature is sufficient, the article's organization is coherent and well-designed, and the conclusions are well-supported, insightful, and have the potential to prompt future research.

My only brief feedback is regarding patient selection:

Although all the pathologies covered by the ICD-10-CM codes are related to TBI, they are arguably quite distinct from one another. Limiting the specific pathologies included in the statistics could decrease the number of patients significantly. However, it may still be worthwhile to investigate whether the most common types of TBI (such as SAH and SDH, as mentioned in the paper) alone yield the same conclusions.

Also, there is a typo in line 104 regarding minor TBI.

Author Response

I would like to thank the reviewer for the invaluable comments. We have added Table 3 to show electrolyte imbalance based on the type of TBI lesion, in which hypermagnesemia and hypercalcemia groups have very low numbers of patients, making it difficult to make a proper comparative analysis. We have mentioned this limitation and also highlighted the need for future studies with adequate sample sizes in each category for a conclusive study. The typo error was corrected and highlighted in yellow in the text.

Reviewer 2 Report

The present study evaluated the predictor role of the initial serum level of calcium and magnesium in hospitalized TBI patients. Hypocalcemia and hypomagnesemia resulted common in hospitalized TBI patients but hypocalcemia was not a significant predictor of mortality while hypermagnesemia was an independent predictor.

 The study is well written and interesting topic.

Despite this:

Statistical Analysis 

Please insert the name software analysis.

Results: 

-       The flow chart, figure 1, just consider calcium and serum level, and the other parameters that showed statistical significance? (see table 5)

-       I suggest to the authors of insert a heat map for visualization data, to help the reader in understanding the results

Discussion:

The author reported that “The bivariate analysis further showed that calcium level was also linked with other clinical variables such as bicarbonate level”, but in discussion is not treated, why?

Just as, lactate data are not discussed, why?

The authors must indicate the novelties and the strength lights of their work

Minor

Line 102-104

Following a head injury, consciousness was evaluated using the Glasgow Coma Scale 102 (GCS), which has a range of 3 to 15 and a severity scale of 0 to 8 for severe, 9 to 12 for moderate, and 1 to 13 for minor head injuries.

GCS range is 3-15 or 0-15? Minor head injuries is from 13 and 15?

Author Response

We would like to thank the reviewer and we have addressed the required comments as following:

The name of software analysis was added

The flow chart considered the prevalence of serum Ca and Mg levels (normal, hyper and hypo) and mortality in each group to provide an overall picture of the study main finding. Table 5 summarizes the actual lab results. Therefore, we think that adding the headmap for visualization data will not add more information

We have commented on the lactate and bicarbonate in the discussion: The bivariate analysis found that hypocalcemia was associated with lower bicarbonate level, this may be due to shared physiological pathways or mechanisms. However, hypomagnesemia was associated with lower lactate. Elevated levels of lactate in the bloodstream can lead to metabolic acidosis and hinder the brain's capacity to control blood flow and sustain typical brain activity. This, in turn, can raise the likelihood of secondary brain injury and mortality.

For the novelty we added that Our findings on the role of magnesium and calcium levels in predicting TBI mortality and pattern of lesions could add to the existing literature and provide valuable insights into the complex interplay between electrolyte imbalances and TBI pattern and outcomes.

The ranges of GCS corrected

Reviewer 3 Report

On my opinion, authors have done a great job. Undoubtedly, the topic is urgent. The article is well written. A large source material for analysis is presented.

However, the purpose of the study is not presented in the introduction. Most of the discussion material should be presented in the introduction, since quite a lot of research groups are working on the problem. There are 176 publications in PubMed for the query "Magnesium and traumatic brain injury" (29 in the last 5 years). For the query "Calcium and traumatic brain injury" there are 410 publications in the last 5 years. It is necessary to present in the introduction what actually prompted the authors to make such analysis, despite the fact that the role of calcium and magnesium ions as predictors of the fatal outcome of traumatic brain injury has been already studied (e.g., doi: 10.3390/nu14194174; doi: 10.4103/ajns.AJNS_232_16,  doi: 10.1089/neu.2007.0277, doi: 10.4103/1793-5482.161171, dOI: 10.1097/TA.0000000000003027).

Having decided on the goal, the authors will be able to discuss novel knowledge they have received in comparison with the known data or build a retrospective analysis - from the past to the present (as it was stated in the title).

Technical notes:

Paragraph from lines 47 to 58 - references are required.

Line 311 - "this study", change the letter "t" to capital.

On the one hand, authors made a good description of the problem, but on the other hand, the purpose of the presented publication is not entirely clear.

Author Response

The purpose of the study is not presented in the introduction:

Reply: We have already mentioned the following in the introduction: The present study evaluated the predictor role of the initial serum level of calcium and magnesium in hospitalized TBI patients. We hypothesized that abnormal electrolytes levels on admission are associated with unfavorable outcomes after traumatic brain injury.

We agree that there are publications addressing the calcium and magnisum in TBI, however, our study described both electrolytes in one paper and we done many analyses to address the impact of the normal , hypo and hyper levels showing the importance of hypermagnesemia on the outcome in contrast to the other laboratory findings

Paragraph from lines 47 to 58 - references are required.:

Reply: Please clarify which sentences need references as the given lines are in the abstract

Line 311 - "this study", change the letter "t" to capital.

Reply: Corrected thanks

On the one hand, authors made a good description of the problem, but on the other hand, the purpose of the presented publication is not entirely clear.

Reply: The present study evaluated the predictor role of the initial serum level of calcium and magnesium in hospitalized TBI patients. We hypothesized that abnormal electrolytes levels on admission are associated with unfavorable outcomes after traumatic brain injury.

Round 2

Reviewer 2 Report

The paper is now ok.

Reviewer 3 Report

This paragraph in the introduction: "The tight regulation of the coagulation cascade, which is essential for maintaining 47 hemostasis, is mostly mediated by calcium ions. In addition to platelet activation, calcium 48 ions also fully activate several other coagulation factors, such as coagulation Factor XIII. 49 Clotting factor IV is a calcium ion that plays an important role in the intrinsic, extrinsic, 50 and common pathways. Calcium is a divalent cation that can exist in several states, in- 51 cluding a free, unbound, physiologically active state as well as an inert state that is at- 52 tached to different proteins. While 55% of total calcium is bound to proteins (i.e., albumin) 53 and citrate, only about 45% of it is physiologically active and resides in the ionized state. 54 Derangements in the total body storage and serum can result from variations in the quan- 55 tities of these proteins in the serum. Recent research has focused on hypocalcemia in 56 trauma patients to improve resuscitation and comprehend the relationship between cal- 57 cium derangements, mortality risk, and transfusion requirements" - references are required. 

On the one hand, authors made a good description of the problem, but on the other hand, the purpose of the presented publication is not entirely clear.